# Alternative splicing across the tree of life

**Rebeca de la Fuente**[1]*, **Wladimiro Díaz-Villanueva**[1,2,3†], **Vicente Arnau**[1,2,3†], **Andres Moya**[1,2,3†]

[1]Foundation for the Promotion of Sanitary and Biomedical Research of the Valencian Community (FISABIO), Valencia, Spain; [2]Institute of Integrative Systems Biology (I2Sysbio), University of Valencia and Spanish National Research Council (CSIC), Valencia, Spain; [3]Center for Biomedical Research in Epidemiology and Public Health Network (CIBEResp), Madrid, Spain

## eLife Assessment

The authors examined the frequency of alternative splicing across prokaryotes and eukaryotes and found that the rate of alternative splicing varies with taxonomic groups and genome coding content. This **solid** work, based on nearly 1500 high-quality genome assemblies, relies on a novel genome-scale metric that enables cross-species comparisons and that quantifies the extent to which coding sequences generate multiple mRNA transcripts via alternative splicing. This timely study provides an **important** basis for improving our general understanding of genome architecture and the evolution of life forms.

**\*For correspondence:**
science.rdelafuente@hotmail.
com

[†]These authors contributed equally to this work

**Competing interest:** The authors declare that no competing interests exist.

**Abstract** There is a growing understanding of how alternative splicing contributes to functional specialization and adaptation, especially in well-studied model organisms. However, its large-scale evolutionary dynamics remain poorly understood. Through a comparative analysis of alternative splicing across 1494 species spanning the entire tree of life, this study integrates numerous lines of prior evidence to provide a unified view of alternative splicing. We propose a novel genome-scale metric designed to support cross-species comparison. Our findings indicate that alternative splicing is highly variable across lineages. While unicellular eukaryotes and prokaryotes display minimal splicing, mammals and birds exhibit the highest levels of alternative splicing. Despite sharing a conserved intron-rich genomic architecture, mammals and birds show considerable interspecies divergence in splicing activity. In contrast, plants display moderate levels of alternative splicing but exhibit high variability of genomic composition. Furthermore, a strong negative correlation is observed between alternative splicing and the proportion of coding content in genes, with the highest levels of alternative splicing observed in genomes containing approximately 50% intergenic DNA.

## Introduction

Alternative splicing is the post-transcriptional mechanism by which different protein isoforms are generated from a single gene by the selective combination of exons and introns (*Rogalska et al., 2024*; *Bush et al., 2017*; *Chen et al., 2012*). Thus, it allows functional diversification of a single gene without increasing the number of genes (*Chen et al., 2014*; *Chen et al., 2012*), though it relies on a highly sophisticated set of splicing factors and regulatory pathways to maintain accuracy. This process is central to organismal complexity, regulating gene expression across different tissues, developmental stages, and environmental conditions. On one side, it is involved in essential biological functions, such as tissue differentiation (*Chen et al., 2014*; *Wright et al., 2022*), neural development, and immune response (*Kim et al., 2007*; *Rogozin et al., 2012*; *Wolf and Koonin, 2013*), where it is highly

conserved to maintain functional integrity and precise gene regulation. However, its weak splice sites allow for greater plasticity, enabling adaptation to different conditions (*Caron et al., 2017*). As a result, it also serves as a major driver of biological innovation, promoting functional diversification and adaptability by expanding transcriptomic and proteomic diversity (*Su et al., 2006*). It has been described as a form of 'evolutionary tinkering,' where gradual accumulation of mutations can introduce or remove splice sites, leading to the production of alternative protein isoforms with modified structural and regulatory properties. Thus, it enables functional adaptations while maintaining essential biological processes (*Mattick, 2009*).

Over the past 1400 million years, alternative splicing rates have steadily increased, particularly within the metazoan lineage, coinciding with the emergence of specialized cell types and tissue-specific gene regulation. Prokaryotes and unicellular eukaryotes exhibit significantly lower alternative splicing rates, supporting the idea that it represents an advanced regulatory feature associated with multicellularity (*Rogozin et al., 2012*). On the other side, vertebrates exhibit significantly higher levels of alternative splicing than invertebrates (*Hinman and Cary, 2017*), and different taxonomic groups have developed distinct splicing strategies. Animals, in particular mammals, birds, and arthropods, display the highest levels of alternative splicing, whereas plants show intermediate splicing levels, suggesting that it evolved independently in this lineage. Notably, *Chen et al., 2014* found no significant difference in alternative splicing levels between birds and mammals, contradicting earlier reports. In contrast to animals, several studies have shown that plants exhibit lower alternative splicing rates but compensate through gene duplication and genome expansion via transposable elements (*Koralewski and Krutovsky, 2011*). In *Koonin, 2006*, authors observed that some large plant genomes, such as maize, exhibit higher alternative splicing rates than smaller genomes like *Arabidopsis*, suggesting a positive correlation between genome complexity and transcriptomic flexibility. In plants, genome expansion frequently occurs through whole-genome duplications and repetitive element accumulation, both of which create additional opportunities for splice site recognition and regulation. Polyploidy, the presence of multiple chromosome sets, is a widespread feature in plant evolution, with approximately 70% of flowering plants having undergone whole-genome duplications. These duplications lead to subfunctionalization, where duplicated genes evolve different splicing isoforms to fulfill distinct functional roles, thereby increasing alternative splicing diversity. Another major factor influencing alternative splicing in plants is the expansion of transposable elements, particularly retrotransposons, which significantly contribute to genome size and structural variation. As a consequence, plant genomes accumulate large non-coding intergenic regions due to expansion of transposable elements, further influencing alternative splicing regulation.

The expansion of non-coding genomic regions has played a key role in the evolutionary development of alternative splicing. The evolution of coding DNA sequences is not uniform across genes; essential genes involved in transcription, translation, and metabolic processes tend to evolve more slowly, while genes related to immune responses and environmental adaptation undergo faster rates of sequence change (*Cavalier-Smith, 2005*). It has been revealed that highly expressed genes evolve at slower rates due to their structural stability and efficient folding (*Koonin, 2009*). Selection pressure against protein misfolding constrains sequence variation, particularly in essential genes, such as those encoding core cellular machinery. As a consequence, the constraints imposed on coding DNA sequences also vary across the major groups, with prokaryotes displaying higher sequence conservation due to compact genomes and high selective pressures. In contrast, multicellular eukaryotes exhibit higher plasticity, reflecting the impact of relaxed selection and functional specialization (*Koonin and Wolf, 2010*). Comparative genomic studies have revealed that while coding DNA sequences are subject to strong functional constraints, non-coding elements exhibit greater evolutionary plasticity, often expanding through genome duplications and transposable element activity (*Ahnert et al., 2008*; *Silva et al., 2020*). Non-coding sequences introduce new regulatory elements that influence alternative splicing and gene expression patterns (*Patil et al., 2014*; *Koonin and Wolf, 2010*; *Wolf and Koonin, 2013*; *Li et al., 2024*). In particular, increased intron length correlates with greater transcriptomic complexity as it is involved in the fine-tuning of gene expression through alternative splicing mechanisms (*Oesterreich et al., 2016*).

Together, these observations highlight the complex interplay between non-coding sequence expansion and splicing regulation, although the precise relationship between genome architecture and alternative splicing dynamics remains to be fully elucidated. This gap motivates the need for

standardized, genome-wide metrics to better understand splicing patterns across species. Advances in high-throughput sequencing technologies have revealed an unprecedented level of transcript diversity, uncovering previously unrecognized splicing events across multiple species and contributing to our knowledge of its regulatory mechanisms (*Shapiro, 2017*). Several studies have developed databases for alternative splicing analysis (*Kim et al., 2007*; *Itoh et al., 2004*). However, while these studies have uncovered important gaps, they remain isolated findings. Most of these databases focus on model organisms, such as human and mouse genomes, and primarily focus on conserved splicing events. In addition, most comparative studies rely on orthology-based methods and multiple sequence alignment. While these techniques are effective for detecting evolutionary conservation in alternative splicing patterns, these methodologies restrict their applicability in large-scale comparative analyses across distantly related taxa. To date, no study has integrated the vast body of knowledge generated. There is still no standardized metric available for cross-species comparisons of alternative splicing. Here, we propose a whole-genome measure, the *alternative splicing ratio* (ASR), which can be computed from high-quality annotation files such as those provided by the NCBI (*Pruitt et al., 2012*). Additionally, we examine methodological biases in the NCBI annotation files, evaluating how factors such as RNA-Seq depth, computational approaches, and tissue diversity influence alternative splicing detection. Thus, we provide a standardized measure that minimizes these biases, providing a reliable framework for cross-species comparisons. This study represents the first large-scale comparative genomic analysis that integrates data from alternative splicing events across more than a thousand species, encompassing the entire tree of life, and offering a comprehensive perspective on their evolutionary relationship.

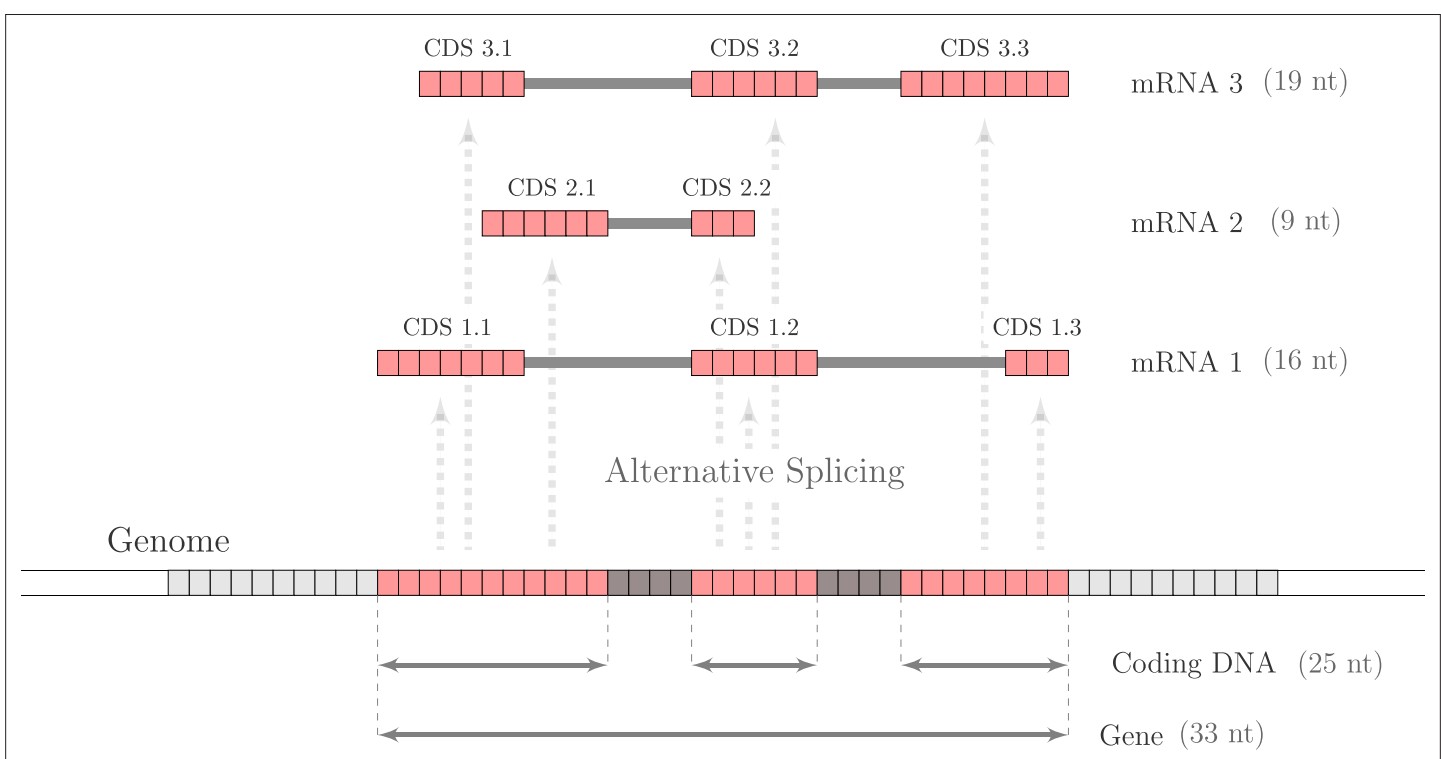

**Figure 1.** Schematic representation of the compositional structure of a gene and alternative splicing. The selective combination of exons and introns in a gene of 33 nucleotides gives rise to three distinct mRNA isoforms: mRNA1 (16 nucleotides), mRNA2 (9 nucleotides), and mRNA3 (19 nucleotides). When these coding DNA sequences (CDSs) are mapped onto the genome, the coding DNA—defined as the DNA sequences that are transcribed into a mRNA—is found to be composed of 25 nucleotides. The ASR is then computed as the ratio of the total number of nucleotides in mRNA isoforms to the number of nucleotides composing coding DNA: $ASR = (16 + 9 + 19)/25 = 1.76$.

## Results

### Genome variables

The compositional structure of genomes is organized into hierarchical domains, reflecting distinct levels of DNA sequence organization. To characterize this structure, we consider three key genomic variables: genome size, gene content, and coding content. Within the genome, the gene content represents the total amount of DNA that forms genes, encompassing both coding and non-coding regions. Coding content, in turn, refers to the amount of DNA within genes that is transcribed into mRNA. Unlike untranslated regions (UTRs) or intronic sequences, coding DNA sequences (CDSs) are composed of sequences that contribute directly to mRNA synthesis.

Alternative splicing is the process by which different combinations of exons and introns are selectively retained or excluded during pre-mRNA processing, resulting in the production of distinct mRNA isoforms. Here, we propose a novel genome-wide metric, the ASR, which quantifies the average number of distinct transcripts generated per coding sequence. The ASR is mathematically defined in 'Methods' and schematically illustrated in *Figure 1*. Although it is computable for single genes, it is extended to the entire genome by mapping every transcribed CDS detected in a species onto its genomic coordinates. Thus, it provides a global quantification of alternative splicing, capturing the extent to which coding DNA sequences are reused across the entire transcriptome. By summarizing this complexity into a single numerical value, it enables large-scale comparative analyses across species, facilitating the study of transcriptomic diversity and evolutionary patterns of splicing regulation. As described in 'Methods,' ASR was computed for 1494 species spanning the entire tree of life. Additionally, to correct for annotation-related biases, we computed a normalized value, ASR*, that accounts for inconsistencies in genome annotations across species. This normalization was necessary to mitigate biases introduced by differences in sequencing depth, tissue diversity, assembly quality, and computational gene prediction, ensuring more accurate cross-species comparisons.

### Variation in alternative splicing among major clades

We performed a comparative analysis to identify variations in alternative splicing across different taxonomic groups. We conducted comparisons of mean values using Welch's ANOVA with Bonferroni correction to control for multiple testing. To further validate our findings, we also conducted a Monte Carlo permutation test, which supported the conclusions (see 'Methods'). Due to the presence of extreme outliers, we conducted the permutation test by comparing both the mean and the median. These two measures of central tendency provide complementary insights into the data distributions, which are non-normal in most cases. The mean, while sensitive to extreme values, gives an overall measure of centrality, whereas the median offers a more robust estimate that is less influenced by

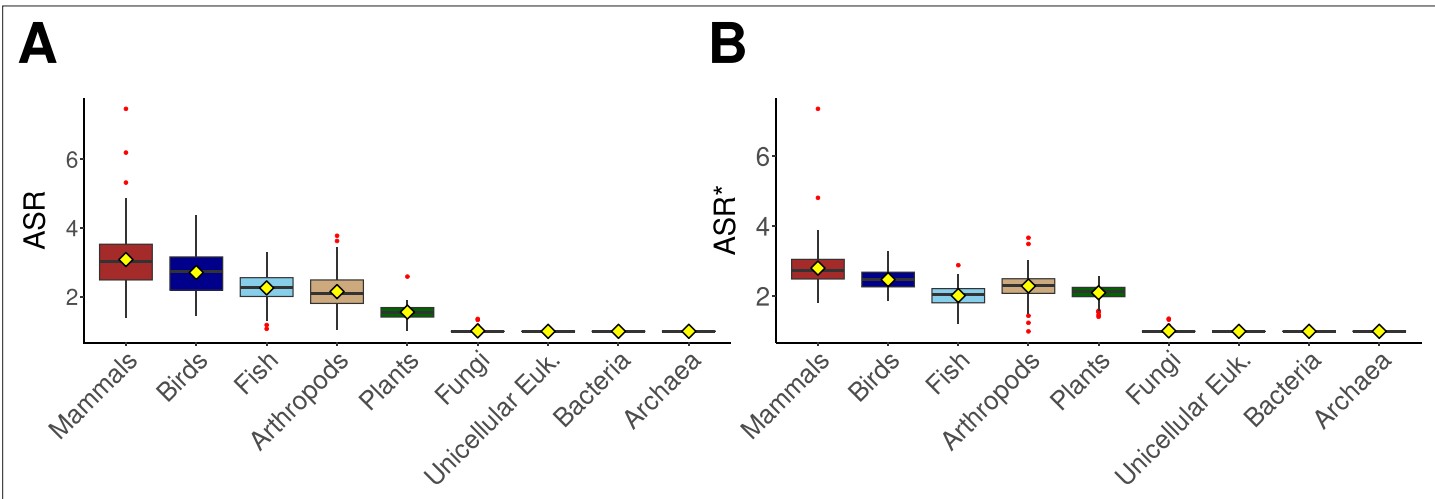

**Figure 2.** Comparative analysis of (**A**) alternative splicing ratio (ASR) and (**B**) normalized ASR (ASR*) distributions across taxonomic groups, including mammals, birds, fish, arthropods, plants, fungi, unicellular eukaryotes, bacteria, and archaea. Box plots represent the median (horizontal line), interquartile range (IQR, box), and whiskers extending to 1.5× IQR. A yellow diamond-shaped point represents the mean, and outliers are shown as individual red points. Colors in the box plots correspond to taxonomic classifications.

**Table 1.** Summary statistics for the percentage of gene content relative to genome size (Gene Content/Genome Size (%)),the percentage of coding relative to gene size (Coding Size / Gene Content (%)),the percentage of coding relative to genome size (Coding Size / Genome Size(%)), the alternative splicing ratio (ASR),and the normalized alternative splicing ratio (ASR*) across different taxonomic groups.

The table includes the mean ($\bar{x}$), the interpercentile range between the 5th and 95th percentiles ($[Q_{0.05}, Q_{0.95}]$), and standard deviation (σ) for each group.

| Group | Gene Content / Genome Size (%) | | | Coding Size / Gene Content (%) | | | Coding Size / Genome Size (%) | | | ASR | | | ASR* | | |
|---|---|---|---|---|---|---|---|---|---|---|---|---|---|---|---|
| | $\bar{x}$ | $[Q_{0.05}, Q_{0.95}]$ | σ | $\bar{x}$ | $[Q_{0.05}, Q_{0.95}]$ | σ | $\bar{x}$ | $[Q_{0.05}, Q_{0.95}]$ | σ | $\bar{x}$ | $[Q_{0.05}, Q_{0.95}]$ | σ | $\bar{x}$ | $[Q_{0.05}, Q_{0.95}]$ | σ |
| Mammals | 44.7 | [35.5,55.6] | 6.27 | 3.05 | [2.47,3.76] | 0.42 | 1.35 | [1.09,1.60] | 0.15 | 3.09 | [1.78,4.60] | 0.92 | 2.81 | [2.11,3.50] | 0.59 |
| Birds | 52.6 | [41.8,62.5] | 6.05 | 4.79 | [4.18,5.53] | 0.45 | 2.5 | [2.21,2.83] | 0.2 | 2.71 | [1.76,3.62] | 0.64 | 2.48 | [1.96,2.96] | 0.31 |
| Fish | 61.9 | [49.4,74.3] | 7.50 | 8.65 | [5.23,13.4] | 2.52 | 5.39 | [2.69,8.57] | 1.77 | 2.26 | [1.55,2.92] | 0.41 | 2.02 | [1.57,2.50] | 0.29 |
| Arthropods | 60.3 | [33.1,79.9] | 14.0 | 11.3 | [2.15,25.1] | 7.15 | 6.92 | [0.91,15.1] | 4.36 | 2.16 | [1.34,3.14] | 0.54 | 2.3 | [1.59,2.90] | 0.39 |
| Plants | 24.7 | [7.04,40.4] | 11.0 | 27.3 | [14.3,38.5] | 7.50 | 6.80 | [1.69,13.6] | 3.68 | 1.56 | [1.30,1.87] | 0.2 | 2.1 | [1.61,2.46] | 0.24 |
| Fungi | 63.2 | [41.2,85.9] | 13.8 | 86.9 | [67.6,99.2] | 11.5 | 55.3 | [33.2,82.7] | 15.4 | 1.01 | [1,1.09] | 0.05 | 1.01 | [1,1.09] | 0.05 |
| Unicellular Eukaryotes | 60.9 | [42.5,88.1] | 16.2 | 88.6 | [69.2,99.9] | 12 | 53.6 | [33.0,76.3] | 14.9 | 1.00 | [1,1.01] | $2.56 \times 10^{-3}$ | 1.00 | [1,1.01] | $2.56 \times 10^{-3}$ |
| Bacteria | 85.7 | [76.8,92.2] | 5.06 | 98.8 | [97.7,99.5] | 0.57 | 84.6 | [75.6,91.2] | 5.07 | 1.00 | [1.00,1.00] | $8.21 \times 10^{-4}$ | 1.00 | [1.00,1.00] | $8.21 \times 10^{-4}$ |
| Archaea | 86.1 | [74.5,92.4] | 5.21 | 99.1 | [98.7,99.4] | 0.22 | 85.3 | [73.9,91.5] | 5.10 | 1.00 | [1.00,1.01] | $1.72 \times 10^{-3}$ | 1.00 | [1.00,1.01] | $1.72 \times 10^{-3}$ |

outliers. Quantitative results of the Monte Carlo permutation test are presented in *Supplementary file 1*.

To explore how alternative splicing varies across the tree of life, we first compared the distribution of ASR and ASR* values across a broad range of taxonomic groups (*Figure 2*). The results revealed significant differences for most comparisons, with only a few exceptions. Overall, unicellular organisms—including archaea, bacteria, fungi, and unicellular eukaryotes—tend to exhibit comparable levels of alternative splicing, with particular similarity observed between unicellular eukaryotes, fungi, and bacteria. Among multicellular taxa, mammals and birds display similar ASR values, as do arthropods and fish. Slight differences were observed in the comparisons when using the normalized ASR value (ASR*). In particular, we found that plants exhibit values comparable to those of fish, despite their distant evolutionary relationship. Similarly, arthropods and birds also display similar ASR* median values. As shown in *Table 1* , the average ASR values indicate that mammals exhibit the highest alternative splicing activity (3.1), followed by birds (2.7), fish and arthropods (2.2), plants (1.6), and unicellular organisms (1), the latter indicating minimal or no alternative splicing. When normalized ASR* values are considered, the ranking among multicellular taxa is maintained, with mammals showing the highest values (2.8), followed by birds (2.5), arthropods (2.3), and both fish and plants with values around 2.

In addition to alternative splicing, we also examined whether certain genomic composition variables differ across major taxonomic groups. Specifically, we focused on the proportion of the genome composed of genes, the proportion of coding DNA sequences within genes, and the overall proportion of coding DNA sequences within the genome. By assessing whether these genomic variables also distinguish taxonomic groups, we can evaluate to what extent variation in alternative splicing values may be influenced by genomic composition. Consistent with this, we found significant differences across most taxonomic groups in both mean and median values of the genomic variables (see *Supplementary file 2*). As reported in *Table 1*, bacteria and archaea exhibit the highest gene content proportions, with genes accounting for approximately 86% of their genomes. This percentage is followed by fish, arthropods, fungi, and unicellular eukaryotes, all of which show similar percentages of approximately 61%, with no significant differences among them. Birds (53%) and mammals (45%) show moderate gene content, whereas plants have the lowest proportion among all groups, at around 25%. On the other side, the coding-to-gene proportion increases progressively across taxa as follows:

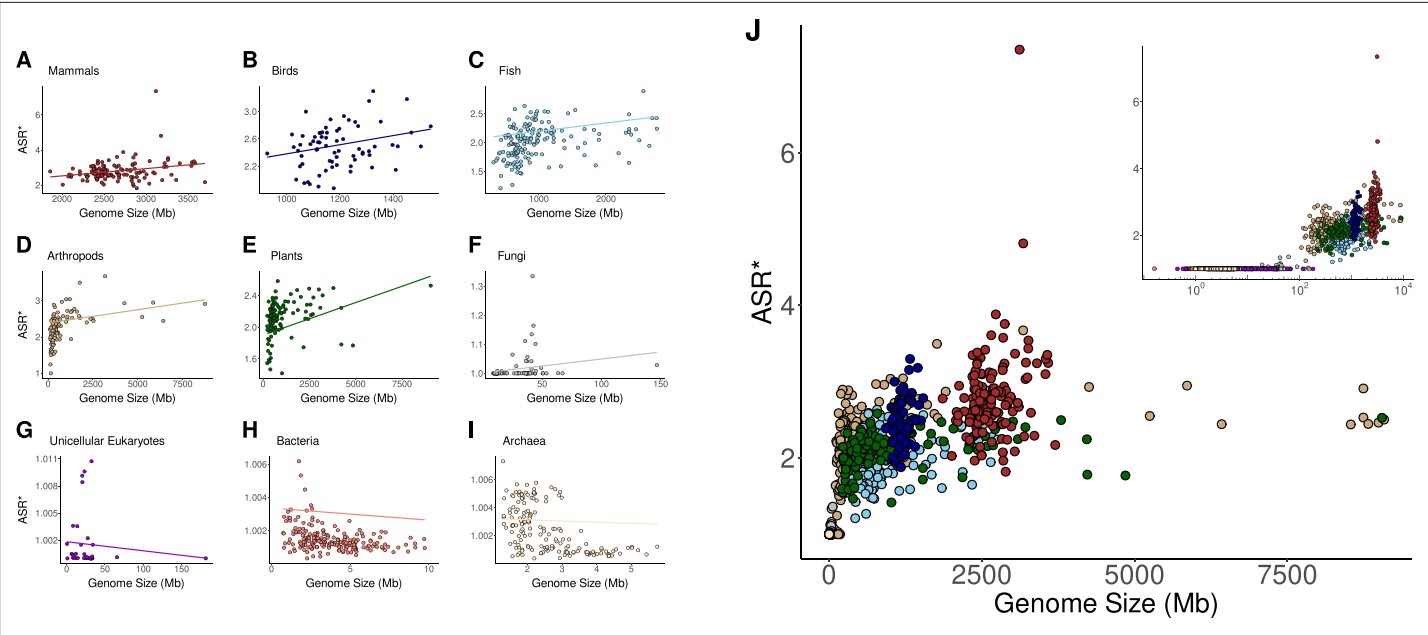

**Figure 3.** Normalized alternative splicing ratio vs genome size across taxonomic groups. (**A–I**) Phylogenetic generalized least squares (PGLS) regression between the genome size and the normalized alternative splicing ratio (ASR*) across different taxonomic groups. Each panel represents a distinct taxonomic group. The regression lines represent the estimated evolutionary relationship between the two variables while accounting for phylogenetic dependence. (**J**) Global relationship across all taxonomic groups. The inset provides a logarithmic representation of the x-axis.

The online version of this article includes the following figure supplement(s) for figure 3:

**Figure supplement 1.** Alternative splicing ratio vs genome size across taxonomic groups.

**Figure supplement 2.** Normalized alternative splicing ratio vs gene content across taxonomic groups.

**Figure supplement 3.** Alternative splicing ratio vs gene content across taxonomic groups.

**Figure supplement 4.** Normalized alternative splicing ratio vs coding DNA across taxonomic groups.

**Figure supplement 5.** Alternative splicing ratio vs coding DNA across taxonomic groups.

mammals (3%), birds (5%), fish (9%), arthropods (11%), plants (27%), fungi and unicellular eukaryotes (88%), and finally, bacteria and archaea, each with approximately 99%.

## Evolutionary patterns of alternative splicing

Understanding genomic expansion patterns and its relationship with alternative splicing across taxa is key to determining whether genome evolution follows universal constraints or taxon-specific trajectories. To systematically address this question, we conducted two complementary analyses. On one side, we performed a phylogenetic generalized least squares (PGLS) to identify possible relationships among genomic percentages and alternative splicing while accounting for phylogenetic dependence (see *Supplementary file 3*). On the other side, we analyzed the relative variability of each genomic variable across taxonomic groups using the coefficient of variation (*Supplementary files 4 and 5*). First, we computed the relative variability of each genomic feature $x$, ($CV_x$), providing a means to quantify the extent of variation within each taxonomic group. Second, we computed variability ratios between pairs of genomic features, ($CV_x/CV_y$), allowing us to compare the variability of one genomic trait to another within and across taxa. A detailed explanation of both methods can be found in the 'Methods. section.

As shown in *Figure 3* and *Figure 3—figure supplement 1*, PGLS analyses revealed no significant correlation between alternative splicing and genome size across any taxonomic group for both ASR and ASR* values. However, in mammals and birds, alternative splicing showed a strong association with both gene and coding content (*Figure 3—figure supplements 2–5*). This association further extends to genomic proportions, particularly the ratio of coding to gene content (*Figure 4* and *Figure 4—figure supplement 1*). These correlations remain consistent for both ASR and ASR*, except

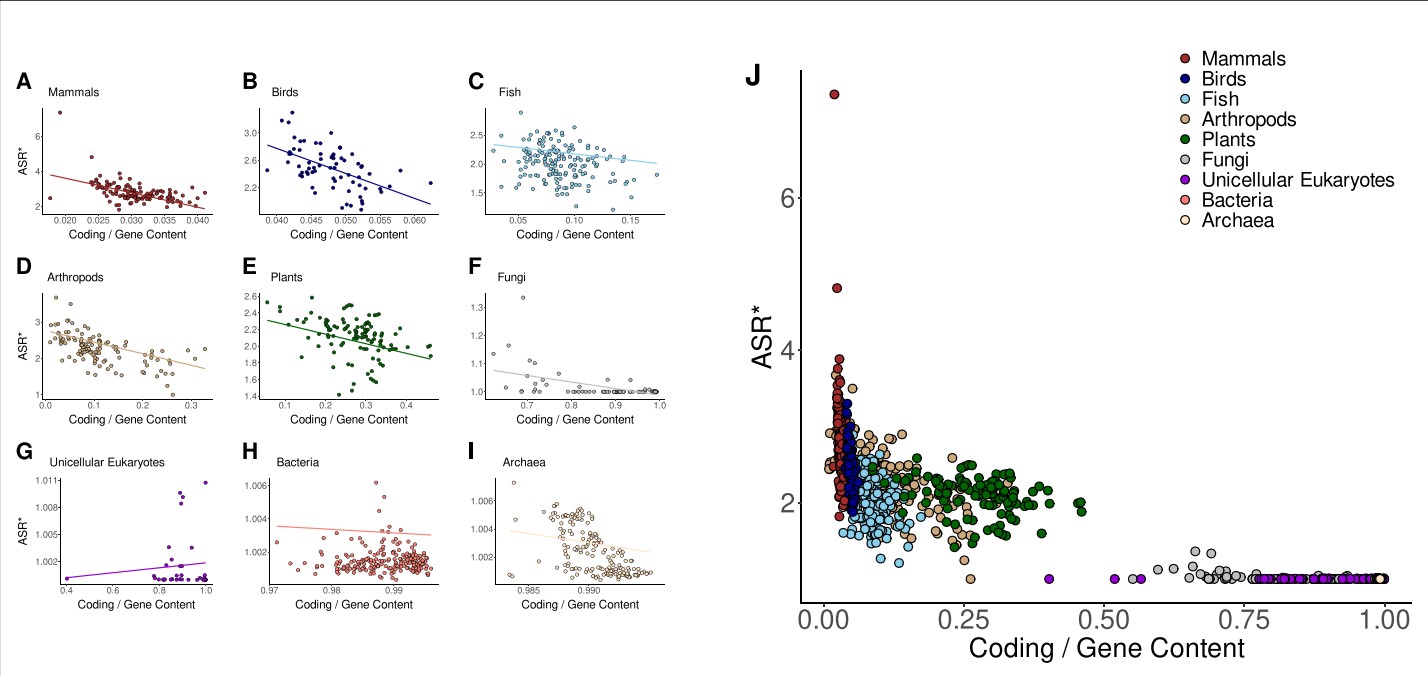

**Figure 4.** Normalized alternative splicing ratio vs the percentage of coding DNA across taxonomic groups. (**A–I**) Phylogenetic generalized least squares (PGLS) regression between the proportion of coding relative to gene content and the normalized alternative splicing ratio (ASR*) across different taxonomic groups. Each panel represents a distinct taxonomic group. The regression lines represent the estimated evolutionary relationship between the two variables while accounting for phylogenetic dependence. (**J**) Global relationship across all taxonomic groups.

The online version of this article includes the following figure supplement(s) for figure 4:

**Figure supplement 1.** Alternative splicing ratio vs the percentage of coding DNA across taxonomic groups.

in the case of coding content, which does not correlate with ASR* in any taxonomic group. These findings indicate that, for mammals and birds, genomes with larger amounts of gene content and coding DNA sequences exhibit higher levels of alternative splicing. When analyzing relative proportions rather than absolute content, we found that alternative splicing (ASR and ASR*) is strongly associated with two key genomic ratios: the genomic fraction occupied by genes and the proportion of coding sequence within genes (*Figures 4 and 5*). In the latter case, we observe a strong negative correlation, supporting the hypothesis that genes with a higher intronic fraction are more likely to give rise to multiple isoforms through alternative splicing.

Notably, in contrast to mammals and birds, neither gene content nor coding content shows an association with alternative splicing (ASR or ASR*) across the other taxonomic groups examined. The only exception is observed in plants, where gene content shows a positive association with ASR*, albeit considerably weaker than the correlation observed in mammals and birds. When considering genomic proportions, however, the analysis reveals correlations in some lineages, although differences are observed between ASR and ASR*. Nonetheless, these associations are generally weaker than those observed in mammals and birds, suggesting that the association between genomic architecture and alternative splicing is stronger in these two groups. In fish, the proportion of genes within the genome is positively associated with both ASR and ASR*. In addition, gene proportion is positively associated with ASR in arthropods and with ASR* in plants. The coding-to-gene ratio appears as a relevant predictor of ASR, correlating with both ASR and ASR* in plants and fungi. Furthermore, its correlation with ASR* is also significant in arthropods and archaea. Another genomic proportion indicative of the genome structure is the coding-to-genome ratio. While ASR shows no significant correlation with this ratio in any taxonomic group, ASR* exhibits a positive correlation in both arthropods and plants. Overall, these findings indicate that the genomic architecture of coding and gene content, and not only their abundance, may play a central role in shaping splicing complexity across most multicellular lineages, particularly in mammals and birds. In these groups, alternative splicing is strongly associated

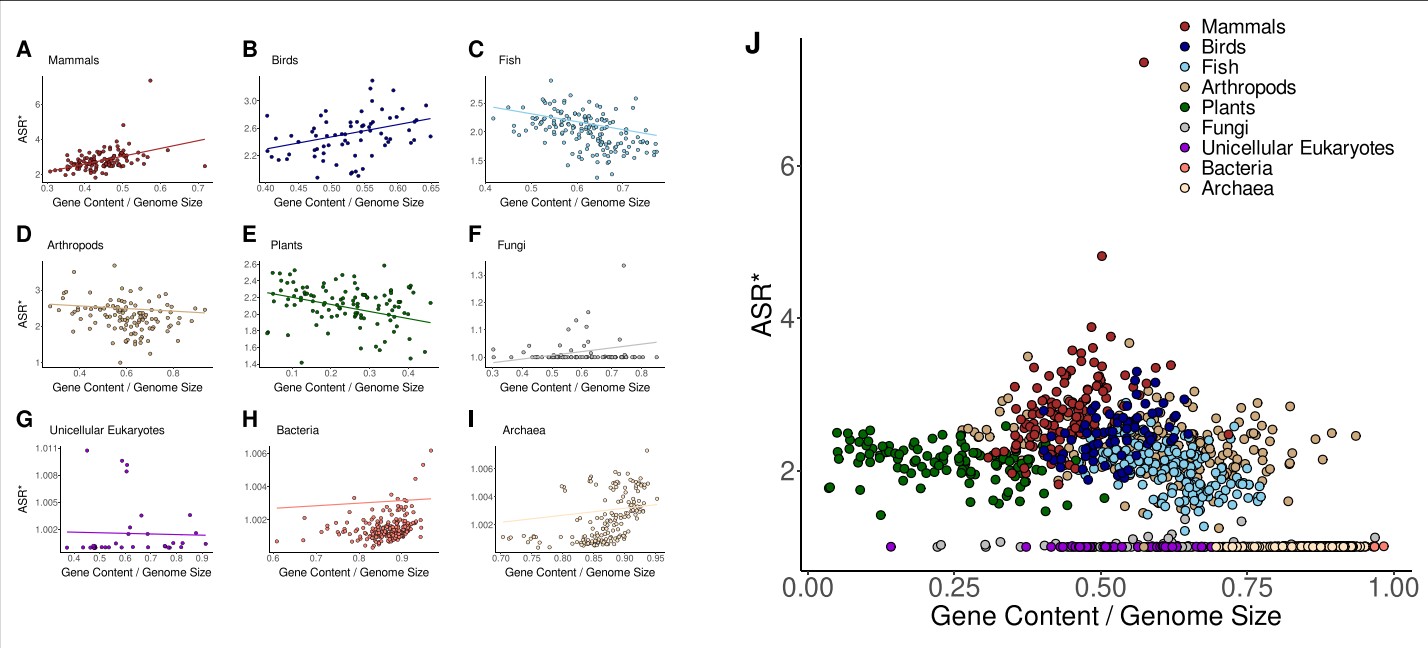

**Figure 5.** Normalized alternative splicing ratio vs the percentage of gene content across taxonomic groups. (**A–I**) Phylogenetic generalized least squares (PGLS) regression between the proportion of gene content relative to genome size and the normalized alternative splicing ratio (ASR*) across different taxonomic groups. Each panel represents a distinct taxonomic group. The regression lines represent the estimated evolutionary relationship between the two variables while accounting for phylogenetic dependence. (**J**) Global relationship across all taxonomic groups.

The online version of this article includes the following figure supplement(s) for figure 5:

**Figure supplement 1.** Alternative splicing ratio vs the percentage of gene content across taxonomic groups.

with intron-rich genes, and the results may reflect a shared, lineage-specific trajectory in the evolution of transcriptomic complexity.

Beyond lineage-specific associations—which may contribute to broader patterns—an overarching trend is observed when examining variability across taxa. We first quantified the relative variability of alternative splicing using the coefficient of variation, which expresses dispersion normalized by the mean, making it independent of scale. Unicellular organisms exhibit negligible variability in alternative splicing, as expected given their minimal or absent splicing activity. Within these groups, only fungi show a slight degree of variation, though still close to zero. In contrast, multicellular taxa display considerable variability in alternative splicing values, with clear differences among groups. Among them, plants consistently exhibit the lowest alternative splicing variability, both in ASR and normalized ASR* measures. Arthropods, fish, and birds show higher variability than plants in ASR; however, after normalization in ASR*, their variation is only slightly higher than plants. Finally, mammals exhibit the highest levels of variability in both ASR and ASR*, indicating a high degree of heterogeneity in splicing activity. Also, as a measure of relative variability, we used the ratio of coefficients of variation to compare how much alternative splicing values vary in relation to genomic features. Specifically, the variability ratio between ASR and the percentage of coding within genes, $\hat{C}V_{ASR}/\hat{C}V_{Coding/Gene}$, follows a progressive trend from unicellular organisms to animals, with mammals and birds exhibiting the highest ratios. These relations are visually reflected in *Figure 4* and *Figure 4—figure supplement 1*. A variability ratio above 1 indicates that alternative splicing is more variable across species within a taxonomic group than the proportion of coding. Notably, only birds and mammals display values exceeding 1 for both ASR and its normalized value ASR*, highlighting a greater degree of interspecific heterogeneity in splicing relative to the proportion of coding composing genes. As a consequence, these lineage-specific patterns of variability in splicing activity, and their strong association with gene architecture, may reflect specific regulatory strategies in mammals and birds. At the opposite end of the spectrum, unicellular organisms exhibit uniformly low variability ratios, indicating that the percentage of coding varies more across species than alternative splicing levels. This pattern is consistent with the near absence of alternative splicing in these groups, which show little to no

variation. Among them, unicellular eukaryotes display the lowest variability ratio, approaching zero, reflecting an almost complete lack of variability in splicing relative to coding content. Between these two opposite trends, fish, plants, and arthropods exhibit intermediate variability ratios, all around 0.5 for both ASR and ASR* values. Although slight differences exist, the decreasing trend from fish to plants and arthropods remains consistent, indicating that these groups display a balanced degree of variation in alternative splicing compared to coding content. This result suggests a continuum between two distinct strategies: one characterized by coding-rich genomes with minimal alternative splicing, as observed in prokaryotes, and another in which small changes in intron-rich gene architectures are coupled with high variations in alternative splicing levels—as seen in mammals and birds.

A phase-like transition in alternative splicing activity is observed at approximately 20 Mb of coding DNA, a value that appears to be a threshold separating unicellular and multicellular organisms. As shown in *Figure 3—figure supplements 4 and 5* , species below this threshold exhibit negligible alternative splicing values. Beyond this threshold, which is exceeded in all multicellular taxa, alternative splicing levels increase. Still, alternative splicing values vary substantially across species, indicating that while a shift toward more complex post-transcriptional regulatory strategies is common among multicellular organisms, the degree of splicing activity is modulated by different factors, including fine-scale differences in gene architecture.

Finally, as previously observed, a significant relationship exists between alternative splicing and the proportion of gene content in most multicellular groups, suggesting that intergenic regions may play a relevant role in shaping splicing dynamics. Otherwise, it may reflect an indirect association with the regulatory architecture of alternative splicing. These patterns are visually represented in *Figure 5* and *Figure 5—figure supplement 1*. The results further indicate that both ASR and ASR* reach their highest values in genomes with approximately 50% intergenic content, supporting the hypothesis that this genomic configuration may favor optimal conditions for alternative splicing. Notably, while mammals and birds exhibit a positive association between ASR* and the proportion of gene content, fish and plants display an inverse pattern, with regression slopes shifting from positive to negative. In mammals, where genes represent less than 50% of the genome, ASR* increases with the percentage of genic content, peaking at 50%. In contrast, fish—whose genomes often exceed this threshold— further gene expansion is no longer positively associated with splicing activity, potentially indicating an evolutionary constraint on genome organization. After examining the correlation between genic proportion and ASR*, we next explored the relationship between the number of genes and ASR. This analysis revealed strong positive correlations in mammals, birds, fish, and arthropods. Altogether, these results suggest the existence of an optimal intergenic composition that enhances alternative splicing.

## Discussion
### A whole-genome approach for quantifying alternative splicing

In this study, we propose a whole-genome measure to quantify alternative splicing, which can be computed using genome annotation files such as those provided by the NCBI Genome Annotation Pipeline. This measure is robust, as we exclusively consider high-quality genomic data from assemblies at the chromosome and whole-genome levels. Since genomes with poor assembly quality are not considered, potential biases in the annotation of alternative isoforms due to low-quality data are minimized. Furthermore, the NCBI pipeline applies the same computational model to annotate both animal and plant genomes, so we also minimize potential biases arising from differences in annotation methodologies between these groups. Our evaluation of methodology-associated biases introduced by the NCBI annotation pipeline reveals that the only significant bias is related to the level of experimental support, with better-supported annotations showing higher alternative splicing estimates. Specifically, the number of CDSs introduced in the pipeline is positively correlated with higher alternative splicing values. As a consequence, this factor represents a potential source of bias. To address this, we apply a normalization to ASR values, effectively correcting for discrepancies introduced by differences in annotation confidence and RNA-Seq coverage across taxa. Furthermore, taxonomic comparisons and lineage-specific patterns in alternative splicing yielded similar results for both ASR and the normalized metric ASR*, reinforcing the validity of ASR as a robust metric for large-scale comparative analyses.

While alternative splicing represents a prominent mechanism of transcriptomic diversification, it should be considered that it constitutes only one component of the broader landscape of gene regulation. Structural and behavioral complexity in organisms arises from a combination of regulatory processes—including transcriptional control, chromatin remodeling, epigenetic modifications, and RNA editing—whose combined interactions ultimately shape phenotypic diversity (*Levine and Tjian, 2003*; *Shlyueva et al., 2014*; *Feschotte, 2008*). In this study, we focus specifically on alternative splicing as a measurable and comparable proxy for regulatory complexity at the genome level. However, we recognize that a more comprehensive analysis of organismal complexity would require the integration of additional layers of regulation and functional data across lineages.

### Evolutionary strategies of transcriptomic diversification

While previous studies have highlighted the role of splice site strength, exon length, and regulatory element density in shaping splicing outcomes (*Bortfeldt et al., 2008*; *Itoh et al., 2004*; *Keren et al., 2010*), our results provide a genomic-level perspective that contextualizes alternative splicing within large-scale patterns. Specifically, we find that alternative splicing activity is not randomly distributed among taxa but follows lineage-specific associations with genome composition, particularly with respect to the proportion of coding and non-coding DNA. In mammals and birds—groups that exhibit the highest alternative splicing values—we observe a strong negative correlation between alternative splicing and the proportion of coding DNA within genes. This indicates that intron-rich gene architectures may promote isoform diversity, consistent with models in which exon skipping predominates and weak splice sites are compensated by dense regulatory motifs (*Kalsotra and Cooper, 2011*). Interestingly, these groups exhibit a highly conserved gene structure, with coding regions representing less

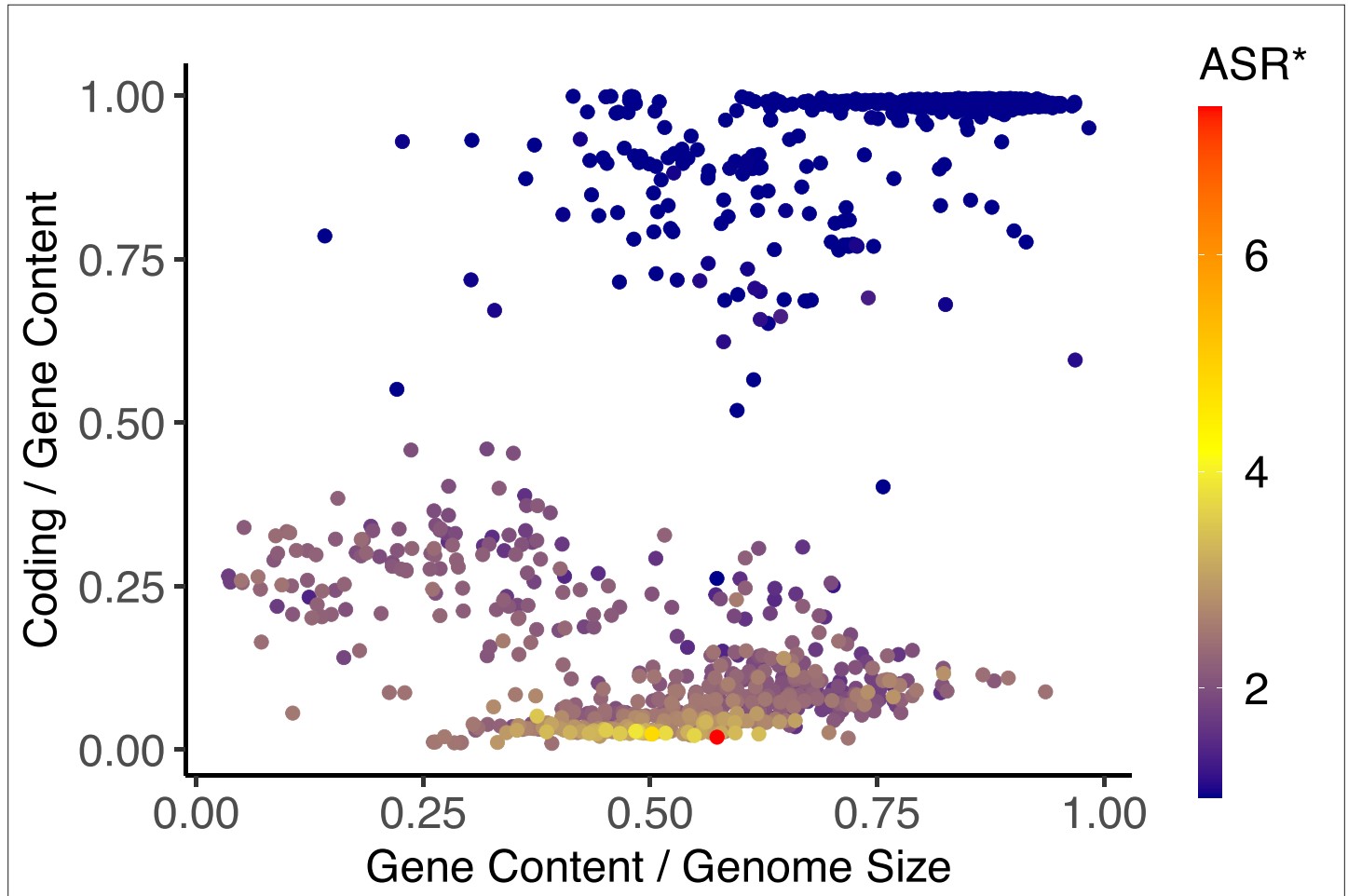

**Figure 6.** Normalized alternative splicing ratio (ASR*) is represented as a color gradient across different genomic profiles. The x-axis spans the gene-to-genome proportion, while the y-axis spans the coding-to-gene proportion.

than 5% of gene length, while alternative splicing levels display considerable variability across species. Thus, within a narrow coding-to-gene compositional range, alternative splicing may be finely regulated, with minor variations in gene structure associated with substantial changes in isoform diversity. This pattern suggests that transcriptomic modularity could be achieved through subtle adjustments in gene architecture, potentially reflecting an evolutionary strategy favoring regulatory refinement over gene duplication. However, future research should investigate the mechanistic basis of this relationship and test whether such fine-tuned structural control indeed underlies shifts in splicing dynamics. In line with models describing alternative splicing as a modular system of transcript generation, this genomic strategy allows functional diversification without the need for coding expansion (*Modrek and Lee, 2002*; *Nilsen and Graveley, 2010*). These findings also align with recent theoretical work proposing that alternative splicing and regulatory complexity increase as coding accumulation slows down, acting as an evolutionary response to structural constraints (*Koonin, 2011*).

Our results reveal a threshold at approximately 20 Mb of coding DNA, separating unicellular from multicellular organisms. Below this coding size, species display a lack of alternative splicing, whereas above it, splicing activity becomes increasingly significant. This observation supports the hypothesis that coding expansion reaches a critical point, beyond which additional genomic innovation depends less on gene duplication and more on regulatory diversification, particularly through alternative splicing (*Lynch, 2007*). Importantly, while coding content increases in multicellular organisms, the relative proportion of coding DNA sequences within genes decreases sharply, especially in mammals and birds. As illustrated in *Figure 6*, once the proportion of coding drops below 50%, alternative splicing emerges significantly, reaching its highest levels in coding-poor organisms. This result is in agreement with previous studies showing that a decrease in coding density is strongly associated with increased splicing activity (*Lynch and Conery, 2000*; *Holland et al., 2017*). As coding expansion slows down, the reliance on alternative splicing intensifies, indicating a shift toward post-transcriptional plasticity as a dominant evolutionary strategy (*Ast, 2004*; *Rogozin et al., 2012*). This shift may be key for shaping broader evolutionary patterns in the relationship between genome composition and transcriptomic complexity across lineages. These findings also support the hypothesis that the slow-down in coding sequence accumulation is intrinsically linked to the emergence of post-transcriptional strategies, such as alternative splicing.

There is a large-scale progressive trend in the variability relationship between coding-to-gene composition and alternative splicing across evolutionary lineages. At one end of the spectrum, prokaryotes exhibit highly compact genomes, with minimal non-coding DNA and negligible alternative splicing activity. In the absence of alternative splicing, these organisms rely on gene duplication as the primary mechanism for functional innovation (*Lynch, 2007*). At the opposite end, mammals and birds show the highest levels of alternative splicing (*Barbosa-Morais et al., 2012*), despite having highly constrained genomic architectures. Between these extremes, plants, arthropods, and fish occupy intermediate positions, with high variability in both coding percentage and splicing levels. This finding supports the existence of two contrasting strategies: one that favors expansion through gene duplication and coding sequence accumulation (as in prokaryotes and some unicellular eukaryotes), and another that maximizes regulatory reuse and complexity through alternative splicing in genomically constrained organisms with high functional plasticity, such as mammals and birds. As a consequence, differences in variability relationships across taxa may be attributed to fundamental shifts in genome expansion strategies. These results support the view that alternative splicing is not merely a means of increasing functional complexity, but also an evolutionary adaptation to structural constraints within the genome.

Our findings also reveal that alternative splicing activity peaks in genomes with approximately 50% of intergenic content, supporting the hypothesis that there exists an optimal genic-to-intergenic balance that maximizes regulatory efficiency. While intergenic regions may contribute to alternative splicing activity, their excessive expansion, as observed in plants, does not result in increased alternative splicing levels. Thus, a threshold of approximately 50% of intergenic content may represent a point beyond which additional non-coding DNA becomes functionally redundant. Thus, rather than promoting continued increases in isoform diversity, expansive intergenic growth may impose architectural limitations that reduce the utility of additional regulatory space. This pattern is consistent with previous work indicating that transcriptomic complexity does not scale linearly with genome size (*Hahn and Wray, 2002*; *McShea, 1996*; *Claverie, 2001*; *Choi et al., 2020*) and may instead

be constrained by saturation effects in regulatory architecture (*Muro et al., 2025*; *Shapiro, 2017*; *Shapiro, 2017*). Future research should explore the mechanisms underlying this apparent intergenic threshold, testing whether regulatory saturation or spatial constraints limit the contribution of additional non-coding regions to alternative splicing complexity.

Plants exhibit a distinct evolutionary strategy compared to animals, particularly in the way non-coding regions have expanded. In contrast to prokaryotes, unicellular eukaryotes, and most animals, where genes constitute over 40% of the genome, plant genomes show a significantly lower gene content, ranging from as little as 7% to a maximum of 40%. This reduction is accompanied by a pronounced expansion of intergenic regions, which varies widely across plant species. Despite this extensive non-coding accumulation, plants display only moderate levels of alternative splicing, suggesting that increased intergenic space does not necessarily enhance transcriptomic complexity (*Chamala et al., 2015*). Intron retention is the predominant mechanism of alternative splicing in plants, whereas exon skipping dominates in animals. Splice sites in plants are also generally weaker and lack the compensatory enrichment of regulatory sequences, such as exonic splicing enhancers, commonly observed in metazoans. These differences imply that alternative splicing in plants may be governed by distinct regulatory constraints, likely shaped by evolutionary processes such as polyploidy, genome duplication, and differential selection on non-coding expansion (*Filichkin et al., 2010*; *Reddy et al., 2013*).

## Evidence for the adaptive role of alternative splicing

One of the major debates regarding alternative splicing is whether it arises as an adaptive trait or as a by-product of reduced selection efficiency (*Lynch, 2007*). Species with smaller effective population size experience weaker purifying selection, allowing the accumulation of slightly deleterious mutations, including those affecting alternative splicing. This 'non-adaptive model' suggests that alternative splicing increases in complex organisms not due to selective advantages but as a consequence of weaker selection pressure in species with small population sizes. However, genomic analyses contradict this view, revealing that alternative splicing is not randomly accumulated but is rather highly conserved in some genes, making it functionally relevant across taxa (*Rogalska et al., 2024*). Advances in long-read RNA sequencing technologies have revealed that while many alternative splicing events may be non-functional or represent splicing errors, a significant subset leads to functionally relevant protein isoforms that are preferentially expressed in specific tissues (*Wright et al., 2022*). Several studies have revealed the conservation of alternative splicing patterns across different animal genomes, such as human, mouse, rat, chicken, fish, and insect genomes, suggesting that splicing regulation plays a fundamental role in evolutionary adaptation (*Kim et al., 2007*; *Holste et al., 2006*). These events are particularly enriched in genes associated with essential biological processes and functionally critical organs, such as the brain, heart, muscles, and testes. This highlights their role in preserving key physiological functions while also contributing to evolutionary adaptation. On the other hand, beyond tissue specialization, alternative splicing is increasingly recognized as a key factor in rapid environmental adaptation, allowing organisms to fine-tune gene expression and modulate protein isoform production in response to external stimuli (*Koonin, 2006*). Stress conditions, such as temperature fluctuations and hypoxia, have been observed to regulate alternative splicing patterns, helping organisms adjust their physiological responses dynamically (*Shapiro, 2017*; *Aravind et al., 2009*). Another example is found in genes associated with the immune system, where alternative splicing modulates immune responses to pathogens, further emphasizing its role in adaptive flexibility. Beyond environmental adaptation, it also plays a major role in biological differentiation at different levels of the biological organization, including transcript diversification (*Chen et al., 2012*; *Nilsen and Graveley, 2010*), cell differentiation (*Fiszbein and Kornblihtt, 2017*; *Fu et al., 2009*), and speciation events, such as morphs (*Steward et al., 2022*; *Grantham and Brisson, 2018*), castes (*Lyko et al., 2010*), or even subspecies (*Harr and Turner, 2010*).

The findings presented in this study support the view that alternative splicing is not simply a by-product of relaxed purifying selection, but instead reflects a finely regulated mechanism linked to genome architecture. While the non-adaptive model states that splicing diversity increases stochastically due to weaker selection in species with small effective population sizes, our results show that alternative splicing levels are strongly associated with genomic architecture, particularly with the proportion of coding DNA sequences within genes. Mammals and birds exhibit highly conserved,

intron-rich gene structures, in which the proportion of coding DNA sequences within genes remains low and stable across species. Despite this structural conservation, these groups display substantial interspecific variability in alternative splicing levels, but this variability is strongly and negatively correlated with the proportion of coding DNA within genes, indicating that even small differences in coding compression are associated with pronounced differences in splicing activity. These results suggest that alternative splicing is highly sensitive to structural variation, rather than accumulating stochastically. It may act as a modular regulatory system that enables transcriptomic diversification without requiring further expansion of coding DNA sequences. Together with prior evidence of tissue-specific conservation and adaptive responses mediated by alternative splicing, our results reinforce its role as a key evolutionary mechanism that integrates structural features of the genome with regulatory plasticity. However, more comprehensive research is needed to determine whether these patterns reflect causal relationships. Future studies should integrate high-quality annotations, functional assays, and comparative analyses to fully disentangle the relative contributions of structural constraints and regulatory complexity.

## Methods

### Alternative splicing ratio: A genome-scale measure

The genome is defined as the sequential arrangement of nucleotides, where the $i$th position of a nucleotide is denoted as $G(i)$. We define $f(i,j)$ as the number of times the nucleotides $G(i)$ and $G(j)$ appear together in a CDS. By way of example, a value of $f(i,j) = 3$ indicates that the nucleotides at positions $i$ and $j$ co-occur in three distinct isoforms. Accordingly, $f(i,i)$ corresponds to the number of different CDSs where nucleotide $G(i)$ is inserted. The *transcription matrix $M$*, with $M_{ij} = f(i,j)$, is a symmetric matrix that encodes the co-occurrence patterns of nucleotides within multiple mRNAs, providing a genotype-phenotype mapping framework for transcriptomic and proteomic diversity. The binary version of the transcription matrix is defined as

$$A_{ij} = \begin{cases} 1, & \text{if } M_{ij} > 0 \\ 0, & \text{if } M_{ij} = 0. \end{cases} \tag{1}$$

The diagonal of this matrix represents the projection of CDSs onto the genome, which maps the nucleotides that constitute the coding DNA. Thus, the ASR is mathematically defined as

$$\rho = \frac{tr(M)}{tr(A)}, \tag{2}$$

where $tr(\cdot)$ denotes the trace of the matrix, that is, the sum of the diagonal elements. It quantifies the extent to which coding DNA sequences can be reutilized through alternative splicing to generate multiple mRNA isoforms. It provides a measure of transcriptomic modularity, reflecting the parallelization of genetic information. Here, we extend the concept of the ASR from individual genes to the whole genome by computing the transcription matrix using all nucleotides that compose genes.

### NCBI RefSeq dataset

Genomic features are computed from high-quality, whole-genome assemblies available in the RefSeq database provided by the NCBI (*National Library of Medicine, 2024*). These assemblies are annotated by the NCBI genome annotation pipeline, which integrates diverse experimental evidence, including RNA-Seq data, ESTs, and protein alignments (*Kitts et al., 2016*). The annotation files are in GFF3 format, and the data is provided in tab-delimited tables, where each row represents an entity (e.g., gene, mRNA, or CDS) and columns contain attributes like unique identifiers, genomic coordinates, and functional annotations. The annotations also include additional attributes such as start and stop codons, and UTRs. Genome size, gene content, and coding region coverage are estimated by looking at the start and end coordinates of each genomic element. Gene content is calculated as the total number of nucleotides falling within annotated gene regions. In contrast, the coding region coverage is determined by summing all nucleotides that belong to at least one CDS.

Each file is organized hierarchically in a parent–child structure that represents the relationship between genes, their transcripts (mRNA), and coding DNA sequences (CDS). At the top level, each gene is uniquely identified and associated with multiple mRNA transcripts, which represent the alternative splicing variants of that gene. These mRNA transcripts, in turn, are linked to one or more CDS, which correspond to the portions of the mRNA that are translated into proteins. The set of CDSs associated with a given mRNA is retained in the spliced mRNA, defining a distinct protein isoform. Hierarchical relationships are preserved through shared identifiers across levels: a gene ID links to its transcripts, and a transcript ID links to its associated CDS entries. This structured format allows for efficient navigation and analysis of the dataset, supporting downstream applications like isoform quantification. Note that we have only considered CDSs with a well-defined parent–child relationship. Pseudogenes and duplicated genes have been excluded for this study.

In this study, we propose a novel genome-level measure of alternative splicing. In annotation files, each isoform is identified by looking at the set of CDSs associated with a given mRNA transcript. Since each gene can produce multiple protein isoforms, the corresponding mRNA transcripts often contain regions with overlapping CDSs. Thus, the metric is calculated as the ratio of the total cumulative length of all annotated CDSs to their length when projected onto the genome. This approach provides novel quantitative insights into estimating alternative splicing activity at the species level.

First, the assembly summary of RefSeq was downloaded, which includes detailed information about the genome annotations available in the database (**NCBI, 2024**). Second, the dataset was subsequently filtered based on the following criteria: (i) only genomes annotated by the NCBI RefSeq annotation pipeline were selected, ensuring consistent and reliable annotations; (ii) assemblies were restricted to those with an assembly level of 'chromosome' or 'complete genome,' which guarantees high-quality data; and (iii) genomes were further filtered to include only those belonging to specific taxonomic groups of interest (see 'Taxonomy assignment'). Finally, the filtered annotation files were downloaded in November of 2024, and the genomic features were computed for each species.

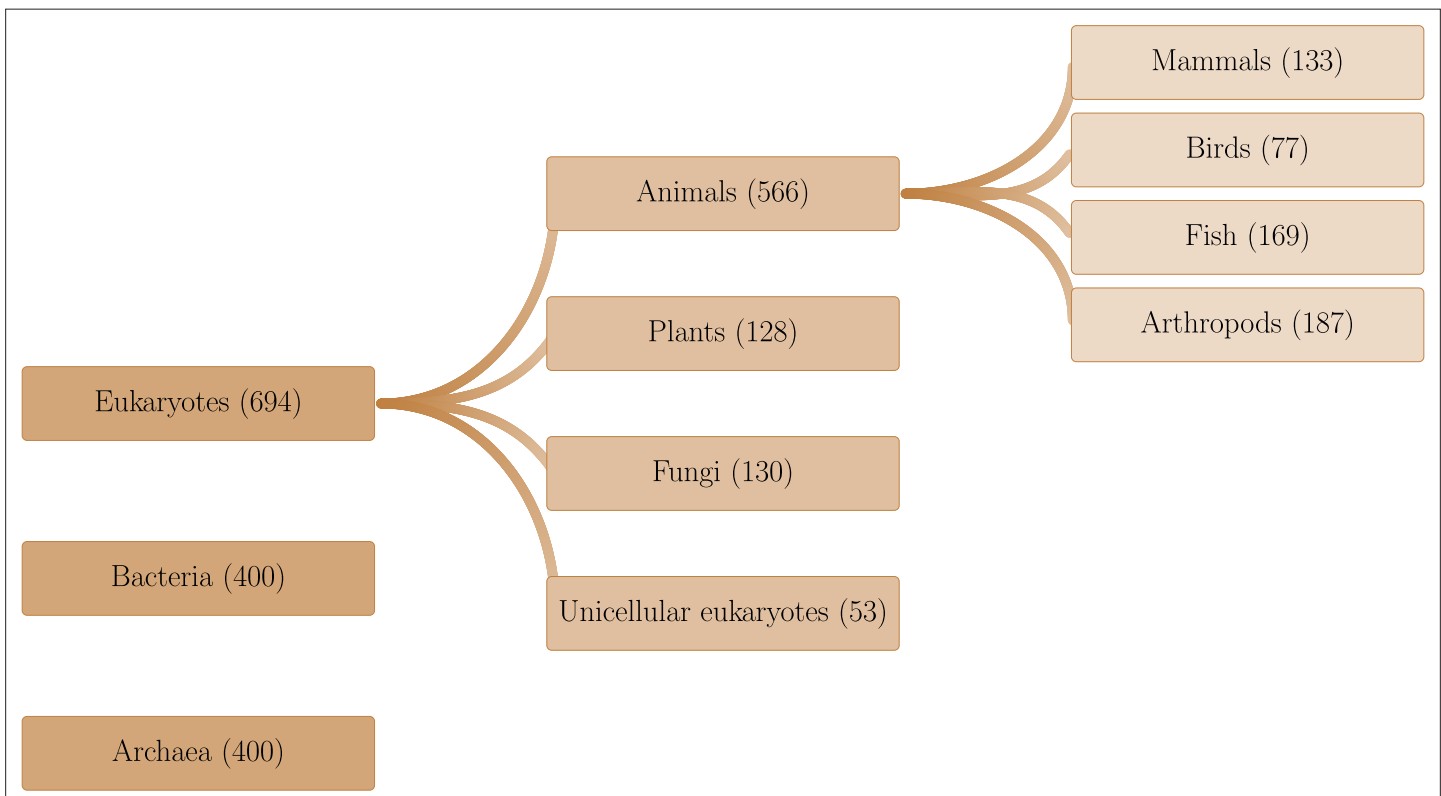

**Figure 7.** Organisms were classified into taxonomic groups based on the hierarchical structure provided by the NCBI Taxonomy Database, using annotation data from NCBI that meet conditions (i) and (ii) described in 'NCBI RefSeq dataset'.

de la Fuente *et al*. eLife 2024;13:RP94802. DOI: https://doi.org/10.7554/eLife.94802

## Taxonomy assignment

We collected annotation files for organisms representing the entire tree of life. However, the sampling process was constrained by the availability of whole-genome annotation files, resulting in a dataset of 694 eukaryotic species. Taxonomy tools from the NCBI (*Sayers et al., 2023*) were utilized to map the phylogenetic relationships among species that meet conditions (i) and (ii) described in the previous section. Next, iTOL was employed for the visualization and schematic representation of these relationships (*Letunic and Bork, 2021*). Taxonomic groups at the levels of kingdom, phylum, and class containing at least 20 distinct species were selected. This approach resulted in the identification of the following major groups: 133 mammals, 77 birds, 169 fish, 187 arthropods, 128 plants, 130 fungi, and 53 unicellular eukaryotes. For the other biological domains, given that the NCBI provides annotations for thousands of species, we randomly selected 400 archaea and 400 bacteria (see *Figure 7*). This approach ensures that the taxonomic groups are well-defined and adequately represented to provide statistical significance for subsequent analyses.

## Methodology-associated biases in the data

In this study, we used assembled genomes annotated by the NCBI genome annotation pipelines. The NCBI Genome Annotation Pipeline is designed for the annotation of both eukaryotic and prokaryotic genomes. On one side, the NCBI Eukaryotic Genome Annotation Pipeline (EGAP) is an evidence-based pipeline for the high-quality annotation of genes, transcripts, and proteins of eukaryotes (*Pruitt et al., 2012*). Each RefSeq assembly generated from the EGAP includes a very detailed annotation report that provides information about the supporting sequencing data and variations in methodology. All eukaryotic genomes annotated to date from the EGAP can be found in the following repository: https://www.ncbi.nlm.nih.gov/refseq/annotation_euk/all/. On the other side, the NCBI Prokaryotic Genome Annotation Pipeline (PGAP) is specifically designed to annotate bacterial and archaeal genomes, focusing on the unique genomic features of these domains (*Tatusova et al., 2016*). Annotations for prokaryotes rely on curated data from the NCBI RefSeq database, including known gene sequences and protein information. It processes assembled genomes, identifying coding DNA sequences (CDSs), ribosomal RNA (rRNA), transfer RNA (tRNA), small RNAs, and other genetic elements. Unlike the eukaryotic pipeline (EGAP), which incorporates complex RNA-Seq alignments and isoform modeling, PGAP emphasizes the detection of essential genes, operons, and conserved elements, as splicing is rare in prokaryotes. The prokaryotic pipeline is optimized for high-throughput processing, ensuring consistent and accurate annotation of thousands of bacterial and archaeal genomes.

While archaeal and bacterial genomes are annotated through the PGAP, annotation files for plants and animals are derived through the EGAP. However, not all RefSeq genomes have been processed through the EGAP pipeline. RefSeq annotations of fungi and unicellular eukaryotes use the annotations as submitted to the International Nucleotide Sequence Database Collaboration, so there is variation in methodology. These organism groups are generally not annotated using RNA-Seq data and have few, if any, alternative isoforms annotated. Nevertheless, this lack of annotation does not imply that these organisms have no alternative splicing. Instead, it reflects limitations in the data and annotation methodologies.

There can be various sources of noise in the annotation of alternative isoforms. Regarding the annotation methodology, we can expect—based on previous evidence—that alternative splicing is a significant mechanism in multicellular organisms. Since the same annotation pipeline is applied to both animals and plants, it is reasonable to assume that genome annotation is not a source of noise for these eukaryotic groups and does not introduce significant biases in the number of alternative isoforms. Moreover, due to the cross-referenced and high-quality data provided by the RefSeq database, we ensure that the annotated features are reliable and comparable across diverse taxa within multicellular eukaryotic groups.

Methodology-associated biases in NCBI annotation files primarily stem from the reliance on the quality and availability of input data. Additionally, the use of reference-based annotation methods can introduce biases by propagating errors or omissions from reference genomes. Variability in sequencing technologies may also impact the accuracy of annotations, potentially leading to biased estimates of alternative splicing events. On the other hand, poor assembly quality can negatively impact the annotation of alternative splicing. However, in this study, we focus exclusively on genomes assembled at

the chromosome or complete genome level, effectively reducing biases caused by assembly quality. Biases in annotation genomes also include the overrepresentation of model organisms, which leads to more accurate annotations for well-studied species. For less-studied species, genome annotations often rely on automated predictions and may lack experimental validation. RNA-Seq depth influences the annotation of alternative splicing events and lncRNAs. Consequently, the amount of annotated alternative isoforms depends on the organism, but that is primarily for humans, mice, and rats, for which we can expect some biases. We have thousands of genome annotations from a variety of species that are not model organisms, so similar biases may exist across many organisms. The amount and diversity of RNA-Seq data used for annotations are highly variable, which can also lead to significant biases. While many organisms are annotated using substantial amounts of RNA-Seq data (billions of reads), this varies considerably across organisms, and tissue diversity is often limited by data availability. Such limitations can affect downstream analyses, reducing the generalizability and reliability of conclusions drawn from these data.

For this study, we have implemented two filtering criteria to reduce methodological biases in the dataset: first, all multicellular organisms were annotated using the EGAP pipeline, which integrates evidence-based computational models; second, only genomes assembled at the chromosome level were included. Despite these measures, potential biases remain, particularly those related to RNA-Seq depth and tissue diversity. To account for potential biases introduced by the annotation process in eukaryotic genomes, we evaluated how the experimental evidence of CDSs affects estimates of alternative splicing. We found that the proportion of coding DNA sequences that are fully supported by experimental evidence is strongly associated with ASR values, whereas annotations mainly based on computational models systematically underestimate splicing complexity. A multivariate regression analysis identified three key predictors of ASR variation: fully supported CDSs, known CDSs, and model-derived CDSs, which are related to each other. Among them, the proportion of fully supported CDSs is the dominant factor. Based on this, we implemented a normalization procedure that adjusts ASR values using a polynomial regression model, effectively removing annotation-related biases. This normalization, which we denote as ASR*, enables more accurate cross-species comparisons while preserving relative differences in splicing levels. However, normalized values should be interpreted as standardized estimates rather than absolute measures, as the correction is empirical and designed to improve comparability across diverse taxonomic groups.

## Monte Carlo permutation tests

Pairwise statistical analysis using Monte Carlo permutation tests was performed to assess differences in genomic variables (e.g., genome size, gene content, coding size, and ASR) across taxonomic groups. Given that some variables present a high number of outliers and, in certain cases, deviate from normality and homoscedasticity—both critical assumptions for many parametric analyses—this approach was employed as a complementary method to Welch's ANOVA, which accounts for heteroscedasticity.

This non-parametric approach evaluates statistical significance by generating a null distribution through random resampling. Specifically, we tested differences using two statistical measures: the mean and the median. First, we computed the observed difference between groups and then compared it to a null distribution generated from 10,000 random permutations, in which group labels were shuffled to simulate the null hypothesis of no difference between groups. The empirical p-value was calculated as the proportion of permuted test statistics greater than or equal to the observed difference. To account for multiple comparisons, we applied the Bonferroni correction, ensuring a stringent control of type I error across all pairwise tests. All analyses in this study confirm that 10,000 permutations provide a robust estimation of the exact test, as increasing the number of permutations to 20,000 yielded statistically equivalent results.

## Comparative genomic analysis

We used the TimeTree platform (timetree.org, *Kumar et al., 2022*) to generate a phylogenetic tree, which provides evolutionary relationships based on common ancestry. The tree was saved in Newick format (.nwk). Since not all species in our dataset (for which we have annotation files) are documented on this platform, the resulting phylogenetic tree includes a subset of species. Specifically, our tree

consists of 170 archaea, 213 bacteria, 35 unicellular eukaryotes, 79 fungi, 110 plants, 116 arthropods, 163 fish, 70 birds, and 122 mammals.

We conducted a PGLS regression to assess the relationships between genomic variables and ASR while accounting for phylogenetic non-independence among species. This method incorporates evolutionary relationships into the statistical model by adjusting for shared ancestry, ensuring that trait correlations are not confounded by phylogenetic structure. To quantify these relationships, we showed the slope ($\beta$), which indicates the direction of the relationship. Positive values indicate an increase in the dependent variable as the independent variable increases, and negative values indicate the opposite. We also showed the p-value, which assesses the statistical significance of this relationship, with lower values ($p < 0.05$) indicating that the observed pattern is unlikely to be due to chance. The adjusted $R^2$, ($R^2_{adj}$), measured the proportion of variance explained by the model while correcting for the number of predictors. Finally, $\lambda$ quantified the phylogenetic signal, with values close to 1 indicating strong phylogenetic dependence and values near 0 reflecting evolutionary independence. These metrics collectively allow us to determine genomic relationships and evaluate whether they are driven by taxon-specific patterns.

We also computed the coefficient of variation ($\hat{C}V$) for each variable, a scale-independent measure of dispersion. The $\hat{C}V$ is defined as the ratio of the standard deviation $s$ to the sample mean $\bar{x}$, expressed as

$$\hat{C}V = \frac{s}{\bar{x}}. \tag{3}$$

This metric quantifies the relative variability of a trait, making it particularly useful for comparing datasets with different units or magnitudes. A higher $\hat{C}V$ indicates greater dispersion relative to the mean, suggesting higher heterogeneity, whereas a lower value implies more constrained variability. By analyzing the $\hat{C}V$ of genomic variables and alternative splicing, we assess the extent of evolutionary variation within and across taxonomic groups. This approach allows us to detect regions of genomic composition that display higher heterogeneity, providing insights into which aspects of genome evolution are more flexible and are subject to stronger evolutionary constraints. Moreover, by comparing these patterns across taxonomic groups, we can assess whether the distribution of genomic heterogeneity follows similar trends across lineages or if taxon-specific factors shape distinct variability regimes. We introduce a metric to quantify these variability relationships, defined as the ratio of the coefficient of variation between different variables, calculated as

$$\hat{C}V_x/\hat{C}V_y, \tag{4}$$

where $\hat{C}V = s/\bar{x}$ is the coefficient of variation of $x$. These ratios provide a scale-independent measure of how the variability of one genomic trait compares to another within and across taxonomic groups.

## Additional information

### Funding

| Funder | Grant reference number | Author |
|---|---|---|
| Generalitat Valenciana | CIPROM/2021/042 | Andres Moya |

The funders had no role in study design, data collection and interpretation, or the decision to submit the work for publication.

### Author contributions

Rebeca de la Fuente, Conceptualization, Software, Formal analysis, Investigation, Visualization, Methodology, Writing – original draft; Wladimiro Díaz-Villanueva, Vicente Arnau, Conceptualization, Supervision, Validation, Methodology, Writing – review and editing; Andres Moya, Conceptualization, Supervision, Funding acquisition, Validation, Methodology, Project administration, Writing – review and editing

## Author ORCIDs
Rebeca de la Fuente https://orcid.org/0000-0003-1536-5689
Wladimiro Díaz-Villanueva https://orcid.org/0000-0002-8970-4367
Vicente Arnau https://orcid.org/0000-0002-1388-6141
Andres Moya https://orcid.org/0000-0002-2867-1119

Reviewer #2 (Public review): https://doi.org/10.7554/eLife.94802.3.sa1
Reviewer #3 (Public review): https://doi.org/10.7554/eLife.94802.3.sa2
Reviewer #4 (Public review): https://doi.org/10.7554/eLife.94802.3.sa3
Author response https://doi.org/10.7554/eLife.94802.3.sa4

# Additional files

### Supplementary files
MDAR checklist

Supplementary file 1. Comparisons of alternative splicing ratio across taxonomic groups.

Supplementary file 2. Comparisons of genomic variables across taxonomic groups.

Supplementary file 3. PGLS analyses.

Supplementary file 4. Coefficient of variation across taxonomic groups.

Supplementary file 5. Relative variability among genomic features.

### Data availability
Data has been downloaded from the NCBI database (*Kitts et al., 2016*). The list of species under study, code, and the datasets generated in this study are available in Zenodo, at https://doi.org/10.5281/zenodo.15189630.

The following dataset was generated:

| Author(s) | Year | Dataset title | Dataset URL | Database and Identifier |
|---|---|---|---|---|
| de la Fuente R | 2025 | sciencerdelafuente/AlternativeSplicing: AltSp | https://doi.org/10.5281/zenodo.15189630 | Zenodo, 10.5281/zenodo.15189630 |

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
