## [Editor Report · eLife Assessment]

The authors examined the frequency of alternative splicing across prokaryotes and eukaryotes and found that the rate of alternative splicing varies with taxonomic groups and genome coding content. This **solid** work, based on nearly 1500 high-quality genome assemblies, relies on a novel genome-scale metric that enables cross-species comparisons and that quantifies the extent to which coding sequences generate multiple mRNA transcripts via alternative splicing. This timely study provides an **important** basis for improving our general understanding of genome architecture and the evolution of life forms.

---

## [Referee Report · Reviewer #2 (Public review)]

Summary:

In this contribution, the authors investigate the degree of alternative splicing across the evolutionary tree, and identify a trend of increasing alternative splicing as you move from the base of the tree (here, only prokaryotes are considered) towards the tips of the tree. In particular, the authors investigate how the degree of alternative splicing (roughly speaking, the number of different proteins made from a single ORF (open reading frame) via alternative splicing) relates to three genomic variables: the genome size, the gene content (meaning the fraction of the genome composed of ORFs), and finally, the coding percentage of ORFs, meaning the ratio between exons and total DNA in the ORF.

The revised manuscript addresses the problems identified in the first round of reviews and now serves as a guide to understand how alternative splicing has evolved within different phyla, as opposed to making unsubstantiated claims about overall trends.

---

## [Referee Report · Reviewer #3 (Public review)]

Summary:

In "Alternative Splicing Across the Tree of Life: A Comparative Study," the authors use rich annotation features from nearly 1,500 high-quality NCBI genome assemblies to develop a novel genome-scale metric, the Alternative Splicing Ratio, that quantifies the extent to which coding sequences generate multiple mRNA transcripts via alternative splicing (AS). This standardized metric enables cross-species comparisons and reveals clear phylogenetic patterns: minimal AS in prokaryotes and unicellular eukaryotes, moderate AS in plants, and high AS in mammals and birds. The study finds a strong negative correlation between AS and coding content, with genomes containing approximately 50% intergenic DNA exhibiting the highest AS activity. By integrating diverse lines of prior evidence, the study offers a cohesive evolutionary framework for understanding how alternative splicing varies and evolves across the tree of life.

Strengths:

By studying alternative splicing patterns across the tree of life, the authors systematically address an important yet historically understudied driver of functional diversity, complexity, and evolutionary innovation. This manuscript makes a valuable contribution by leveraging standardized, publicly available genome annotations to perform a global survey of transcriptional diversity, revealing lineage-specific patterns and evolutionary correlates. The authors have done an admirable job in this revised version, thoroughly addressing prior reviewer comments. The updated manuscript includes more rigorous statistical analyses, careful consideration of potential methodological biases, expanded discussion of regulatory mechanisms, and acknowledgment of non-adaptive alternatives. Overall, the work presents an intriguing view of how alternative splicing may serve as a flexible evolutionary strategy, particularly in lineages with limited capacity for coding expansion (e.g., via gene duplication). Notably, the identification of genome size and genic coding fraction thresholds (~20 Mb and ~50%, respectively) as tipping points for increased splicing activity adds conceptual depth and potential generalizability.

Weaknesses:

While the manuscript offers a broad comparative view of alternative splicing, its central message becomes diffuse in the revised version. The focus of the study is unclear, and the manuscript comes across as largely descriptive without a well-articulated hypothesis or explanatory evolutionary model. Although the discussion gestures toward adaptive and non-adaptive mechanisms, these interpretations are not developed early or prominently enough to anchor the reader. The negative correlation between alternative splicing and coding content is compelling, but the biological significance of this pattern remains ambiguous: it is unclear whether it reflects functional constraint, genome organization, or annotation bias. This uncertainty weakens the manuscript's broader evolutionary inferences.

Sections of the Introduction, particularly lines 72-90, lack cohesion and logical flow, shifting abruptly between topics without a clear structure. A more effective approach may involve separating discussions of coding and non-coding sequence evolution to clarify their distinct contributions to splicing complexity. Furthermore, some interpretive claims lack nuance. For example, the assertion that splicing in plants "evolved independently" seems overstated given the available evidence, and the citation regarding slower evolution of highly expressed genes overlooks counterexamples from the immunity and reproductive gene literature.

Presentation of the results is occasionally vague. For instance, stating "we conducted comparisons of mean values" (line 146) without specifying the metric undercuts interpretability. The authors should clarify whether these comparisons refer to the Alternative Splicing Ratio or another measure. Additionally, the lack of correlation between splicing and coding region fraction in prokaryotes may reflect a statistical power issue, particularly given their limited number of annotated isoforms, rather than a biological absence of pattern.

Finally, the assessment of annotation-related bias warrants greater methodological clarity. The authors note that annotations with stronger experimental support yield higher splicing estimates, yet the normalization strategy for variation in transcriptomic sampling (e.g., tissue breadth vs sequencing depth) is insufficiently described. As these factors can significantly influence splicing estimates, a more rigorous treatment is essential. While the authors rightly acknowledge that splicing represents only one layer of regulatory complexity, the manuscript would benefit from a more integrated consideration of additional dimensions, such as 3D genome architecture, e.g., the potential role of topologically associating domains in constraining splicing variation.

---

## [Referee Report · Reviewer #4 (Public review)]

The manuscript reports on a large-scale study correlating genomic architecture with splicing complexity over almost 1,500 species. We still know relatively little about alternative splicing functional consequences and evolution, and thus, the study is relevant and timely. The methodology relies on annotations from NCBI for high-quality genomes and a main metric proposed by the authors and named Alternative Splicing Ratio (ASR). It quantifies the level of redundancy of each coding nucleotide in the annotated isoforms.

According to the authors' response to the first reviewers' comments, the present version of the manuscript seems to be a profoundly revised version compared to the original submission. I did not have access to the reviewers' comments.

Although the study addresses an important question and the authors have visibly made an important effort to make their claims more statistically robust, I have a number of major concerns regarding the methodology and its presentation.

(1) A large part of the manuscript is speculative and vague. For instance, the Discussion is very long (almost longer than the Results section) and the items discussed are sometimes not in direct connection with the present work. I would suggest merging the last 2 paragraphs, for instance, since the before last paragraph is essentially a review of the literature without direct connection to the present work.

(2) The Methods section lacks clarity and precision. A large part is devoted to explaining the biases in the data without any reference or quantification. The definition of ASR is very confusing. It is first defined in equation 2, with a different name, and then again in the next subsection from a different perspective on lines 512-518. Why build matrices of co-occurrences if these are, in practice, never used? It seems the authors exploit only the trace. A major revision, if I understood correctly, was the correction/normalisation of the ASR metric. This normalisation is not explained. The authors argue that they will write another paper about it, I do not think this is acceptable for the publication of the present manuscript. Furthermore, there is no information about the technical details of the implementation: which packages did the authors use?

(3) Could the authors motivate why they do not directly focus on the MC permutation test? They motivate the use of permutations because the data contains extreme outliers and are non normal in most cases. Hence, it seems the Welch's ANOVA is not adapted. "To further validate our findings, we also conducted

148 a Monte Carlo permutation test, which supported the conclusions (see Methods)." Where is the comparison shown? I did not see any report of the results for the non-permuted version of the Welch's ANOVA.

(4) What are the assumptions for the Phylogenetic Generalized Least Squares? Which evolution model was chosen and why? What is the impact of changing the model? Could the authors define more precisely (e.g. with equations) what is lambda? Is it estimated or fixed?

(5) I think the authors could improve their account of recent literature on the topic. For instance, the paper https://doi.org/10.7554/eLife.93629.3, published in the same journal last year, should be discussed. It perfectly fits in the scope of the subsection "Evidence for the adaptive role of alternative splicing". Methods and findings reported in https://doi.org/10.1186/s13059-021-02441-9 and https://www.genome.org/cgi/doi/10.1101/gr.274696.120 directly concern the assessment of AS evolutionary conservation across long evolutionary times and/or across many species. These aspects are mentioned in the introduction on p.3. but without pointing to such works. Can we really qualify a work published in 2011 as "recent" (line 348-350)?

The generated data and codes are available on Zenodo, which is a good point for reproducibility and knowledge sharing with the community.

---

## [Author Response]

The following is the authors’ response to the original reviews.

**Reviewer #1**

Methodological biases in annotation and sequencing methods

We acknowledge the reviewer’s concern regarding methodological heterogeneity in genome annotations, particularly regarding the use of CDS annotations derived from public databases. In response, we have properly addressed the potential sources of bias in estimating alternative splicing (AS) across such a broad taxonomic range.

Given the methodological challenges encountered in this study, we have undertaken an in-depth analysis of the biases associated with genome annotations and their impact on large-scale estimates of alternative splicing. This effort has resulted in the development of a comprehensive framework for quantifying, modeling, and correcting such biases, which we believe will be of interest to the broader genomics community. We are currently preparing a separate manuscript dedicated to this methodological aspect, which we intend to submit for publication in the near future.

To account for these biases, we performed a statistical evaluation of annotation quality by examining the relationship between ASR values and multiple features of the NCBI annotation pipeline, including both technical and biological variables. Specifically, we analyzed a set of metadata descriptors related to: (i) genome assembly quality (e.g., Contig N50, Scaffold N50, number of gaps, gap length, contig/scaffold count), (ii) the amount and diversity of experimental evidence used in annotation (e.g., number of RNA-Seq reads, number of tissues, number of experimental runs, number of proteins and transcripts, including those derived from *Homo sapiens*), and (iii) the nature of the annotated coding sequences (e.g., total number of CDSs, percentage of CDSs supported by experimental evidence, proportion of known CDSs, percentage of CDSs derived from ab initio predictions).

This comprehensive analysis revealed that the strongest bias affecting ASR values is associated with the proportion of fully supported CDSs, which showed a strong positive correlation with observed splicing levels. In contrast, the percentage of CDSs relying on ab initio models showed a negative correlation, indicating that computational predictions tend to underestimate splicing complexity. Based on these findings, we implemented a polynomial normalization model using the percentage of fully supported CDSs as the main predictor of annotation bias. The resulting normalized metric, ASR^∗^, corrects for annotation-related variability while preserving biologically meaningful variation.

We further verified the robustness of this correction by comparing the main results of our study using both the raw ASR and the normalized ASR^*^ across all analyses. The qualitative and quantitative consistency of results obtained with both metrics demonstrates that our findings are not an artifact of methodological bias and validates the reliability of our approach.

Conceptual and Statistical Framework

Our aim was not to investigate specific regulatory mechanisms of alternative splicing, but rather to explore large-scale statistical patterns across the tree of life using a newly defined metric—the Alternative Splicing Ratio (ASR)—that enables genome-wide comparisons of splicing complexity across species. To clarify the conceptual framework, we have revised the manuscript to explicitly state our assumptions, objectives, and the scope of our conclusions. The ASR metric is now briefly introduced in the Results section, with a more detailed mathematical formulation included in the Methods section.

From a methodological standpoint, we have expanded the manuscript to better support the comparative framework through additional statistical analyses. In particular, we now include:

• Monte Carlo permutation tests to assess pairwise differences in splicing and genomic variables across taxonomic groups, which are robust to non-normality and heteroscedasticity in the data.

• Welch’s ANOVA with Bonferroni correction, which accounts for unequal variances when comparing group means.

• Phylogenetic Generalized Least Squares (PGLS) regression, which explicitly models phylogenetic non-independence between species and allows us to infer lineage-specific associations between genomic composition and alternative splicing.

• Coefficient of variation analysis, used to evaluate the relative variability of splicing and genomic traits across groups in a scale-independent manner.

• Variability ratio metrics, designed to compare the dispersion of splicing values relative to genomic features, thereby quantifying trends in regulatory plasticity versus structural constraints.

All methods are thoroughly described in the revised Methods section, and their application is presented in the Results section.

Functional vs. non-functional nature of AS events

We have included a new discussion paragraph addressing the ongoing debate regarding the functionality of alternative splicing and a possible non-adaptive explanation for the patterns observed. While many previous studies suggest that a considerable fraction of AS events might represent splicing noise or non-functional isoforms, our intention is not to adopt this view uncritically. Instead, we cite recent literature to provide a more nuanced interpretation, recognizing both the potential adaptive value and the uncertainty surrounding the functional relevance of many AS events. Thus, rather than assuming that all observed alternative splicing events are adaptive or biologically meaningful, we now emphasize that many patterns may emerge from other processes, such as those associated to genomic constraints.

Terminology and Result Interpretation

The manuscript has been thoroughly revised to improve both the scientific language and the conceptual framing. We have removed inappropriate terminology such as “higher/lower organisms” and “highly evolved”. Also, we have reinterpreted the results. As part of this process, the manuscript has been substantially rewritten to focus on the most meaningful findings. Ultimately, we have retained only those results that specifically concern broad-scale patterns of alternative splicing across taxa, which are now presented with greater clarity and methodological rigor.

**Reviewer #2**

Gene Regulatory Complexity Beyond Splicing Mechanisms

While alternative splicing represents a prominent mechanism of transcriptomic diversification, we agree with the reviewer that it constitutes only one component of the broader landscape of gene regulation. Structural and behavioral complexity in organisms arises from a combination of regulatory processes, and our study focuses specifically on alternative splicing as a measurable proxy within this multifactorial system. To clarify this point, we have added a paragraph in the Discussion section, where we explicitly contextualize alternative splicing within the wider regulatory architecture. In that paragraph, we discuss additional mechanisms that contribute to phenotypic complexity—such as transcriptional control, chromatin remodeling, epigenetic modifications, and RNA editing—citing key literature.

Alternative Splicing Measure and Methodology

While we agree that alternative splicing is not a definitive measure of organismal complexity, we argue that it remains a meaningful proxy for transcriptomic and regulatory diversification, especially when analyzed at large phylogenetic scale. In this version of the manuscript, our goal was not to equate alternative splicing with biological complexity, but rather to quantify its patterns across lineages and evaluate its relationship with genome structure. This point is now explicitly stated in both the Introduction and Discussion.

We also recognize the limitations associated with the use of coding sequence (CDS) annotations from public databases such as NCBI RefSeq. To address this concern, we have conducted a detailed analysis of the potential biases introduced by heterogeneous annotation quality, sequencing depth, and computational prediction, as previously addressed in our response to Reviewer #1.

In response to concerns about unsupported statements, we have completely rewritten the manuscript to ensure that all claims are now explicitly supported by data and grounded in up-to-date scientific literature. We have reformulated speculative statements, removed inappropriate generalizations, and improved the logical flow of the arguments throughout the text. In summary, we have strengthened both the conceptual framework and the methodological foundation of the study, while maintaining a cautious interpretation of the results.

Trends of Alternative Splicing

To address the reviewer’s concern, we have revised the interpretation of trends as used in our analysis. In this study, we define a trend not as a strict directional progression or a linear trajectory across all species, but rather as a broad statistical pattern observable in the relative distribution and variability of alternative splicing across major taxonomic groups. We do not claim that this pattern reflects a universal adaptive pathway. Instead, we interpret it as a signal of differences in regulatory strategies associated to the genome architecture. To avoid misinterpretation, we have rephrased several sentences in the manuscript and explicitly emphasized the variability within groups, and the lack of significant correlations in certain clades.

Inconsistent statistics

The discrepancies pointed out were due to differences between mean and median-based analyses. These have been clarified and consistently reported in the revised manuscript. Error bars, p-values, and a supplementary table summarizing all tests are now included. Furthremore, we have no removed any species from our dataset.